

# Sourdough starters exhibit similar succession patterns but develop flour-specific climax communities

Erin A. McKenney[1,2], Lauren M. Nichols[1], Samuel Alvarado[3,4], Shannon Hardy[5], Kristen Kemp[6], Rachael Polmanteer[7], April Shoemaker[8] and Robert R. Dunn[1]

[1] Department of Applied Ecology, North Carolina State University, Raleigh, North Carolina, United States
[2] North Carolina Museum of Natural Sciences, Raleigh, North Carolina, United States
[3] Department of Biology, University of West Florida, Pensacola, Florida, United States
[4] Biotechnology Program, North Carolina State University, Biotechnology-based Sequencing-based Undergraduate Research Experience (BITSURE), Raleigh, North Carolina, United States
[5] The Exploris School, Raleigh, North Carolina, United States
[6] Moore Square Middle School, Raleigh, North Carolina, United States
[7] River Bend Middle School, Raleigh, North Carolina, United States
[8] Ligon Middle School, Raleigh, North Carolina, United States

Corresponding author
Erin A. McKenney,
eamckenn@ncsu.edu

## ABSTRACT

The microbial fermentation behind sourdough bread is among our oldest technologies, yet there are many opportunities for sourdough science to learn from traditional bakers. We analyzed 16S rRNA sequences in R to assess the bacterial community structure and performance of 40 starters grown from 10 types of flour over 14 days, and identified six distinct stages of succession. At each stage, bacterial taxa correlate with determinants of bread quality including pH, rise, and aromatic profile. Day 1 starter cultures were dominated by microorganisms commonly associated with plants and flour, and by aromas similar to toasted grain/cereal. Bacterial diversity peaked from days 2–6 as taxa shifted from opportunistic/generalist bacteria associated with flour inputs, toward specialized climax bacterial communities (days 10–14) characterized by acid-tolerant taxa and fruity ($p < 3.03e-03$), sour ($p < 1.60e-01$), and fermented ($p < 1.47e-05$) aromas. This collection of traits changes predictably through time, regardless of flour type, highlighting patterns of bacterial constraints and dynamics that are conserved across systems and scales. Yet, while sourdough climax communities exhibit similar markers of maturity (*i.e.*, pH ≤ 4 and enriched in *Lactobacillus* (mean abundance 48.1%), *Pediococcus* (mean abundance 22.7%), and/or *Gluconobacter* (mean abundance 19.1%)), we also detected specific taxa and aromas associated with each type of flour. Our results address important ecological questions about the relationship between community structure and starter performance, and may enable bakers to deliberately select for specific sourdough starter and bread characteristics.

## INTRODUCTION

Bread has played a crucial role in human civilizations and cultures (*Sicard & Legras, 2011*) for at least 14,000 years (*Arranz-Otaegui et al., 2018*). Until the late 1800s, most bread would have been produced using a sourdough culture containing a variety of bacteria and yeasts with diverse origins (*Lahue et al., 2020*; *Bigey et al., 2020*) but similar favored traits (*Ercolini et al., 2013*; *Minervini et al., 2014*; *Harth, Van Kerrebroeck & De Vuyst, 2016*; *Gobbetti et al., 2016*; *Landis et al., 2021*). Sourdough production decreased as commercial baker's yeast became widely available, but bakers are increasingly interested in the microbial ecology that contributes to traditional bread-making practices. In the kitchen and home, bakers and consumers appreciate sourdough bread as an artisanal, prebiotic (*Poutanen, Flander & Katina, 2009*), low-glycemic index food (*D'Alessandro & De Pergola, 2014*). Even before commercial yeast shortages during the COVID-19 pandemic (*Bakalis et al., 2020*), making sourdough bread without commercial yeast was gaining popularity among artisan bakers, home bakers, and large manufacturers of commercial sourdough bread (*Minervini et al., 2012*; *Reale et al., 2019*; *Urien et al., 2019*; *Comasio et al., 2020*).

In the lab, sourdough provides a simple, tractable system for the study of ecological questions as well as a compelling hands-on tool for citizen science-based education (http://studentsdiscover.org/lesson/sourdough-for-science/). In ecology, ideal model systems tend to be relatively low in diversity (tens or hundreds of species, not thousands), easily manipulated, and amenable to detailed experiments on particular species pairs (*Wolfe & Dutton, 2015*; *Chappell & Fukami, 2018*). In the classroom, model systems need to be achievable with cheap, readily available materials; relevant to students' daily lives; and aligned (or alignable) with regional education standards. Sourdough starters fit all these criteria; yet most lab-based sourdough research to date neither reflects nor relates to widespread artisanal or home-baking practices (*Calvert et al., 2021*). Here we explore the potential of sourdough as a model ecological system and model educational system by studying a classic ecological phenomenon, succession, which refers to the sequential replacement of species following a disturbance (*Pickett, Collins & Armesto, 1987*)—in our case, the mixture of flour and water.

The canonical sourdough starter is superficially very simple. Similar proportions of flour and water are mixed, and the mixture is colonized by environmental microbes (*Ercolini et al., 2013*; *Minervini et al., 2014*; *Butters, 2018*; *Bigey et al., 2020*). Flour (*Zhang, Oh & Liu, 2017*), the storage vessel, and the dwelling where the starter is kept (*Barberán et al., 2015*) are potential sources of sourdough microbes, while the mixing utensils and the baker's hands facilitate microbial dispersal (*Reese et al., 2020*). These microbes metabolize the available nutrients (especially sugars) to produce different fermentation products, which facilitate additional changes in community structure through three known phases (*Ercolini et al., 2013*; *Minervini et al., 2014*; *Harth, Van Kerrebroeck & De Vuyst, 2016*; *Gobbetti et al., 2016*). In the first phase, bacteria in the flour metabolize sugars to produce lactic acid, which decreases the starter pH below the tolerance of most food-spoilage organisms but favors the growth of lactic acid bacteria (LAB, *e.g.*, *Lactobacillus*, *Pediococcus*, *Weissella*) and acid-tolerant yeasts, which metabolize sugars to produce

carbon dioxide and aromatic compounds (*Ercolini et al., 2013*; *Minervini et al., 2014*; *Harth, Van Kerrebroeck & De Vuyst, 2016*; *Gobbetti et al., 2016*). LAB increase in abundance during the competitive second phase of sourdough succession and produce allelopathic metabolites that contribute to the prolonged third-phase stability of sourdough starters. For example, LAB and yeasts prevent further colonization by producing acids and carbon dioxide (which disfavors obligate anaerobes), respectively—as well as alcohols, aldehydes, and other volatile organic compounds. These microbial products also lend bread its desirable qualities: acids impart tangy flavors, impact texture, and increase shelf life; carbon dioxide makes the bread rise; and aromatic compounds increase flavor complexity. LAB, and the genus *Lactobacillus* in particular, are especially recognized for their contribution to flavor development through lactic and acetic acid production, proteolysis, and the production of aromatic compounds (*Corsetti & Settanni, 2007*). In many ways, the microbial change that occurs in sourdough starters represents a laboratory (or kitchen) model of successional dynamics akin to those in ponds, old fields or other habitats.

In the last decade or so, researchers have noted that the three-phase sourdough succession process proceeds similarly in wheat and non-wheat flours, but on different timelines (*Sterr, Weiss & Schmidt, 2009*; *Weckx et al., 2010*; *Rodríguez et al., 2016*; *Bender et al., 2018*). The succession of bacterial species and their consequences in different cereals are therefore likely to deviate from our understanding of the "standard case". Gluten-free cereals are commonly used in spontaneous lactic fermented foods (*Oyewole, 1997*), but are not widely used in sourdough applications (though see (*Rodríguez et al., 2016*; *Bender et al., 2018*))—although bakers are interested in the potential application of these gluten free cereals to sourdough baking.

From the perspective of bakers, sourdough starters have reached a "mature" climax community when pH and rise have stabilized and the starter can be reliably used to make bread, in 7 to 14 days (*Reinhart, 2001*; *Forkish, 2012*). If we are able to understand how flour type affects succession in sourdough, while working in a classroom environment, we have the potential to advance our ecological understanding of microbial succession, provide a framework through which teachers and students can contribute additional insights in the future, and inform best practices for bakers including guidance as to when a starter grown from a particular flour type has reached its final successional state and is ready to use. To achieve these interdisciplinary goals, researchers at North Carolina State University partnered with middle school teachers to study the spontaneous succession process in sourdough starters grown from 10 different flour types. Specifically, we assessed the impact of flour type on bacterial community structure (characterized *via* next generation sequencing), starter performance (*i.e.*, production of acids, $CO_2$, and aromatic compounds), and the number of days it took for starters to reach maturity (*i.e.*, performance and composition characteristic of a climax community which directly affect bread characteristics).

*Gänzle & Ripari (2016)* recently suggested that dispersal processes are disproportionately important in young sourdough starters, whereas niche-based processes are more important in older, repeatedly backslopped sourdough starters. If this is the case,

we expect to detect high variation among early successional starters (<7 days old) that is unrelated to flour type. Conversely, we predict that the niche-based processes in mature starters (>7 days old) may manifest in two forms. First, we hypothesize that sourdough starters made with different flour types will become predictably different from each other as they mature from 0–14 days, reflecting the specific bacterial taxa that are best able to compete for the different resources in each flour. Second, we hypothesize that the metabolites produced by bacteria present in the starters at different successional stages will lead to predictable changes in both the starter environment and community structure, reflecting more ancient niche differences among bacterial taxa (*e.g.*, the ability to produce or tolerate acids). Finally, we predict that community differences throughout succession should manifest functional differences among the starters, specifically their aromas.

## MATERIALS AND METHODS

All sourdough production methods used in this research were developed to mimic artisanal and home bakers' practices. As such, our protocols are based on online recipes (https://www.kingarthurbaking.com/recipes/sourdough-starter-recipe), utilize locally available ingredients (flour and distilled water) and easily accessible equipment, and are scaled to generate minimal waste. The following activities were completed for evaluation purposes only and as a result, the North Carolina State University IRB determined that they were not regulated by 45 CFR 46 and IRB approval was not required. No personal information was used to contribute to generalizable knowledge.

### Flour selection

Five researchers grew a total of 40 starters from 10 different flour types: four replicates each of five flours containing gluten—unbleached all-purpose flour (processed from *Triticum aestivum*), red turkey wheat (*Triticum aestivum*), rye (*Secale cereale*), emmer (*Triticum dicoccon*), and einkorn (*Triticum monococcum*)—and five gluten-free flours—teff (*Eragostis tef*), millet (*Eleusine coracana*), sorghum (*Sorghum* spp), buckwheat (*Fagopyrum esculentum*), and amaranth (*Aramanthus caudatus*). Cereal grains are small, hard seeds without a fruit layer, grown as crops and harvested to be dried and either fed to livestock; boiled and consumed as a porridge or pilaf; or milled into flour for baking. While most cereals are grasses (*i.e.*, wheat, rye, millet, sorghum, teff), the seeds of other plant families (*i.e.*, buckwheat, amaranth) are also described as "whole grains". The grains used in this project represent four glutenous and five gluten free cereal lineages that together span more than 527 million years across the angiosperm phylogeny. The all-purpose, red turkey wheat, emmer, einkorn, rye, and millet flours were milled at Boulted Bread Bakery (Raleigh, NC, USA), while the teff, sorghum, buckwheat, and amaranth flours were purchased from commercial vendors online.

### Growing and measuring sourdough starters

In studying sourdough starters and other traditional fermentations, there is a tradeoff between using approaches that best match standard laboratory protocol and those that best match actual baking practices. Because we aim to generate results that can be directly

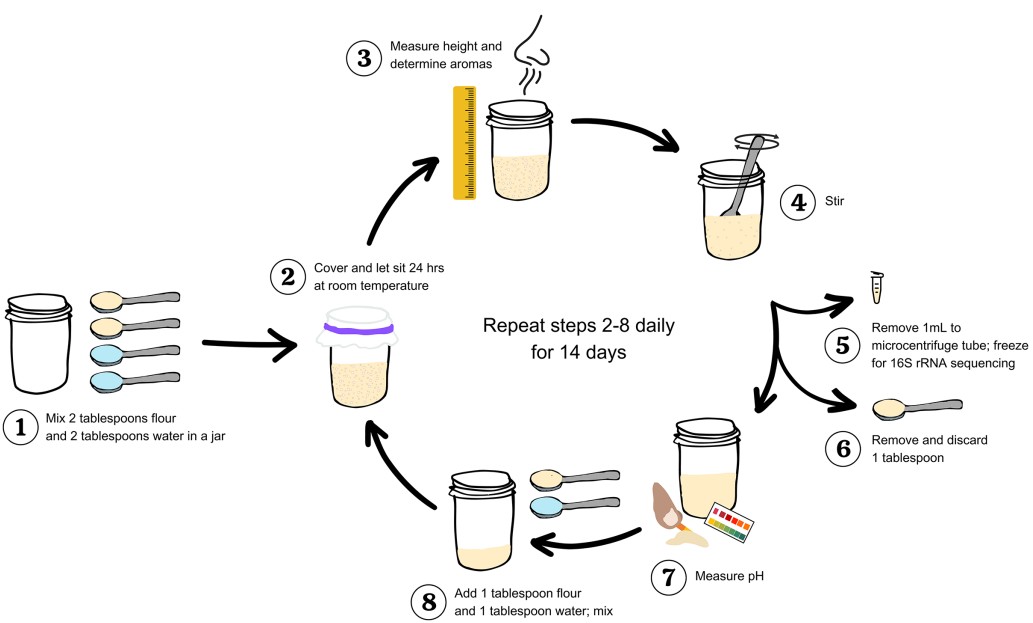

**Figure 1** **Workflow to study and measure a sourdough starter for 14 days.** Step 1: Mix two tablespoons flour and two tablespoons water in a clean jar. Step 2: Cover and let sit 24 h at room temperature. Step 3: Measure maximum height (in centimeters) and determine aromas. Step 4: stir. Step 5: Remove 1 mL starter to a microcentrifuge tube; freeze at −20 °C for 16S rRNA sequencing. Step 6: Remove and discard one tablespoon. Step 7: Measure pH. Step 8: Add one tablespoon flour and one tablespoon water to the starter remaining in the jar; mix. Repeat steps 2–8 daily for 14 days.

applied by bakers, we focused on an approach that is in line with standard baker/bakery practices. We have illustrated our protocol in Fig. 1. On day 0, two level tablespoons of flour and two tablespoons of distilled water were mixed in a half pint wide mouth glass jar with a sterilized spoon. All spoons were washed with soap and water, and wiped with 70% ethanol immediately before mixing. A paper towel was fastened over the mouth of the jar with a rubber band to prevent large particles or insects from settling into the jar while still allowing environmental microbes to colonize the flour-water mixture. The initial height of the flour and water mixture was measured in cm. The jar was labeled with the flour type and replicate number, and left to ferment at room temperature. After 24 h, we measured the maximum height of the starter in cm (*i.e.*, the highest point to which the starter had risen during the preceding 24 h). This point of maximum height can be identified by the residue left on the side of the jar, which resembles a "high tide" mark in cases where an active starter has subsequently deflated by the time of measurement. The height measurements provide a proxy for quantifying carbon dioxide production by each starter community, and changes in community behavior (*i.e.*, function) over time. While sourdough starters grown from different flours may have different gas holding capacities, measuring change in height over time offers insight to gas production for starters grown from the same flour type *via* baseline comparison.

After measuring height, we removed the paper towel and described the smell(s) of the starter using free association, without the aid of a standardized word bank. Each researcher

recorded the aromas produced by every individual starter, each day (1–14). We performed text analysis to classify individual descriptors into super-categories (*i.e.*, earthy, fermented, fruit, grain/cereal, sour, toasted, and other). While sensory assessments may provide subjective interpretation of bread volatiles (*Meignen et al., 2001*), volatile information still correlates with sensory evaluations (*Heenan et al., 2009*), making them useful in the initial description of sourdough aromas.

After recording the aromas, we mixed the starter and transferred 1 mL starter to a sterile, labeled 2-mL microcentrifuge tube, which was stored at −20 °C for DNA sequencing (see *DNA sequencing*). Next, we removed one tablespoon of sourdough (the discard). and measured the pH using short-range (0–6, 0.5 interval) Hydrion Brilliant Paper (MicroEssential Laboratory, Brooklyn, NY, USA). Finally, we added one tablespoon fresh flour and one tablespoon water to the remaining starter and mixed thoroughly with the spoon. These measuring and refreshing steps were repeated once a day (at 24-h intervals) for 14 days. Conventional wisdom holds that a sourdough starter is mature and ready for baking after 14 feedings, although some bakers and instructions for growing a starter from scratch suggest that the starter is mature after only 10 days. After completing data collection, we also baked breads from the starters we grew for this project for an informal tasting, both to celebrate the completion of the project and to verify that all starters successfully leavened bread (Fig. S1).

## Characterizing functional stages of microbial succession

We graphed the pH and height of each starter using ggplot2 in R (*R Core Team, 2019*) to identify distinct stages of succession across time and flour types with regard to starter performance (*i.e.*, production of acid, $CO_2$, and aromas). We used Kruskal-Wallis and Pairwise Wilcoxon rank-sum tests with Bonferroni correction to identify significant differences in pH and height associated with time (*i.e.*, days elapsed) and flour type. We focus on days 1, 2, and 3 to determine the impact of flour type on early succession dynamics and the potential importance of priority effects on community membership and starter performance at maturity. We include day 6 to represent intermediate succession, and days 10 and 14 to acknowledge two common lengths of time by which bakers assume that starters to have reached maturity. We binned aromas by category (*i.e.*, grain/cereal, toasted, nutty, veggie, fruit, fermented, fermented dairy, sour, earthy, bodily smells, chemical, meat) and quantified (presence/absence) whether each aroma category was detected during each sensory assessment).

## DNA sequencing

We collected three 1-mL samples of each flour type (*n* = 30) and three 1-mL samples of distilled water to represent the resource inputs to each starter on day zero. We also collected a daily 1-mL aliquot from each starter after measuring pH, as part of the discard. All aliquots were stored at −20 °C. After analyzing the pH and height data from all starters to identify functionally significant stages in succession, we elected to sequence starter samples collected on days 1, 2, 3, 6, 10, and 14 (*n* = 240). Together with the flour and water aliquots, a total of 273 samples were shipped on dry ice for DNA extraction, PCR

amplification, and Illumina multiplexed sequencing of the bacterial 16S v4 region on the MiSeq platform, using the 515f (AATGATACGGCGACCACCGAGATCTACACGCT TATGGTAATT GT GTGCCAGCMGCCGCGGTAA) and 806r (CAAGCAGAAGACGGCATACGAGAT XXXXXXXXXXXX AGTCAGTCAG CC GGACTACHVGGGTWTCTAAT) primers for bacteria at the Fierer Microbial Community Sequencing Lab (Boulder, CO, USA) as previously described (*Oliverio, Bradford & Fierer, 2017*). PCR negative controls were also sequenced to check for contamination. Raw data are available in GenBank (BioProject PRJNA973060, accessions SAMN35108252–SAMN35108524), and the data and code are available in the Dryad database (https://doi.org/10.5061/dryad.bk3j9kdh3).

## Bioinformatic analysis

The Fierer Microbial Community Sequencing Lab demultiplexed and processed raw sequences to produce exact sequence variant (ASV) tables with the DADA2 pipeline, as described at (https://github.com/fiererlab/dada2_fiererlab) and performed previously by *Landis et al. (2021)* to analyze sourdough communities. We used Cutadapt (*Martin, 2011*) to remove sequences with N's, then quality filtered sequences using truncLen = 150 for forward reads and 140 for reverse reads, maxEE = 1, and truncQ = 11. Next we inferred ASVs with the DADA2 algorithm, merged paired-end reads, and removed chimeras and contaminants before assigning taxonomy with the Silva database (*Quast et al., 2013*). We performed all downstream analyses in the R environment (*R Core Team, 2019*); all associated data and scripts are available in the Supplemental Materials. We filtered out reads assigned to either chloroplast or mitochondria from the bacterial taxa table, along with any reads that were unassigned at the phylum level. We reclassified all ASVs classified as *Lactobacillus* species using the "lactotax" tool (http://lactotax.embl.de/wuyts/lactotax/) developed by *Zheng et al. (2020)* to refine taxonomic structure for the clade. For ASVs originally classified as *Lactobacillus* but without species designations, we used BLAST to determine the new genus designations. First we cross-referenced each of the hits that met the 99% threshold with the new taxonomy. If all of the hits for a specific ASV matched a single genus, we assigned that new genus. If the BLAST results did not agree on a single genus, the ASV was assigned taxonomy only at the family level. We then created two datasets for downstream analyses on alpha and beta diversity:

1) We compared all water, flour, and day 1 starters to assess the influence of inputs on early starter communities. This data set was un-rarefied.

2) We removed all water and flour samples to compare changes in starter community composition from day 1–14. This dataset was rarefied to 1,200 bacterial reads per sample and the ASV table was converted to percent relative abundances for downstream analysis. We retained 232 starter samples for downstream analysis.

We first plotted heatmaps and produced nonmetric multidimensional scaling (NMDS) plots with Bray-Curtis distance calculations using the metaMDS function to compare the bacterial contents of day 1 starters to flour and water inputs. We next turned our attention to focus on the rarefied starter dataset. We calculated alpha diversity as ASV richness and

Simpson diversity. We used Kruskal-Wallis tests to measure the effects of days elapsed on alpha diversity and on Lactobacillaceae ASV abundance. As a non-parametric method, the Kruskal-Wallis test is appropriate for comparing microbial membership and other data that do not follow normal distribution (*Kruskal & Wallis, 1952*). We calculated the Pearson correlation coefficient to quantify the relationship between pH and the relative abundance of Lactobacillaceae to determine the strength of the relationship between microbial membership and metabolic function. We calculated beta diversity between starter samples as Bray-Curtis dissimilarity, using the rarified ASV table. We conducted permutational multivariate analysis of variance (PERMANOVA) using the adonis function in the package vegan to test for relationships between flour type and days elapsed on the composition of bacterial communities. Similarly, we used the envfit function in the vegan package to test for relationships between flour type and aromas on bacterial community composition. We produced nonmetric multidimensional scaling plots with Bray-Curtis distance calculations using the metaMDS function, overlaying the centroids (99% confidence interval for standard error) for each day elapsed. We ran a Kruskal-Wallis test with a Bonferroni correction to determine which bacterial taxa and aromas were significantly different between days (across flour types) and between flour types in mature (day 14) starter communities. We plotted the degree (eigenvalue) and direction (eigenvector) of influence of these significant bacterial taxa and aromas on starter communities using envirofit function from the vegan package. Finally, we created heatmaps using plot_ts_heatmap from the mctoolsr package (https://github.com/leffj/mctoolsr/) to visualize all bacterial genera present at ≥0.1% relative abundance for each time point or flour type, and aromas detected in at least 25% (10/40) of starter samples.

## RESULTS

The raw dataset included 1,219,763 forward reads and 1,215,774 reverse reads, with 665,156 joined reads remaining after quality control.

### Characterizing functional stages of succession

We identified six functionally distinct succession stages across flour types based on starter pH (Kruskal-Wallis, $p < 2.2e{-}16$) and starter height (Kruskal-Wallis, $p < 2.2e{-}16$) (Fig. 2 and Table 1). Notably, all starters rose: the height dynamics we observed were robust (Fig. 2C) and similar (Fig. 2D) across flour types. On day 1, starters showed little to no change in either pH or height compared to day 0. On day 2, pH dropped (Pairwise Wilcoxon Rank Sum, $p < 3.8e{-}09$) while height increased (Pairwise Wilcoxon Rank Sum, $p < 0.00355$). pH continued to fall on day 3, (Pairwise Wilcoxon Rank sum test, $p < 0.00112$), after which starters stabilized at a pH of 3.6; we detected no significant changes in pH throughout the following days of the experiment (Wilcoxon Rank Sum Test, $p = 1.0$). By contrast, height continued to rise significantly through day 6 (Pairwise Wilcoxon Rank Sum Test comparison between days 2 and 6, $p < 0.00112$) and day 10 (Pairwise Wilcoxon Rank Sum Test comparison between days 6 and 10, $p < 0.00477$). By day 14, both pH and height were maintained, suggesting that communities were functionally stable.

**Table 1 Characteristics of six functionally distinct stages of succession characterized across sourdough starters grown from 10 different types of flour.** Kruskal-Wallis (K-W) tests were performed to determine which measures (pH, height), bacterial taxa, and aromas differed significantly between days. Here we list the taxa comprising ≥10% of the community and any aromas that were identified in ≥10 starters, in descending order, to characterize each stage; the prevalence of all taxa and aromas are available in Fig. 3. K-W entries indicate Kruskal-Wallis results for pH, height, bacterial membership, and aromatic profiles. Days with the same letter are not significantly different, while days with different letters are significantly different.

| Functional stage | 1 | 2 | 3 | 4 | 5 | 6 |
|---|---|---|---|---|---|---|
| Days elapsed | 1 | 2 | 3 | 6 | 10 | 14 |
| Average pH ∓ SE | 5.31 ∓ 0.09 | 4.2 ∓ 0.09 | 3.67 ∓ 0.08 | 3.68 ∓ 0.06 | 3.66 ∓ 0.08 | 3.54 ∓ 0.06 |
| pH K-W | a | b | c | c | c | c |
| Average height (cm) ∓ SE | 2.3 ∓ 0.04 | 2.75 ∓ 0.11 | 2.91 ∓ 0.12 | 3.27 ∓ 0.1 | 3.92 ∓ 0.16 | 3.78 ∓ 0.17 |
| Height K-W | a | b | bc | cd | e | de |
| Dominant bacteria | Enterobacteriaceae unkn, Pantoaea, Klebsiella, Pseudomonas, Erwinia | Enterobacteriaceae unkn, Pseudomonas, Weissella, Klebsiella | Enterobacteriaceae unkn, Weissella, Pseudomonas, Klebsiella | Latilactobacillus, Weissella | Latilactobacillus, Pediococcus, Gluconobacter, Levilactobacillus | Pediococcus, Gluconobacter, Levilactobacillus, Latilactobacillus |
| Bacteria K-W | a | b | b | c | d | d |
| Dominant aromas | Toasted, grain.cereal | Grain.cereal, sour | Sour, earthy, grain. cereal | Sour, grain. cereal, fruit | Earthy, sour, toasted | Fruit, sour, fermented |
| Aromas K-W | a | b | bc | cd | e | f |

## Bacterial succession

As predicted, the composition of bacterial taxa in starters on day 1 was driven by taxa associated with flour and inputs (Fig. S2). For example, *Pantoea* and an unknown genus within Enterobacteriaceae dominated both flour (33.1% and 20.3%, respectively) and day 1 starter samples (23.8% and 26.7%, respectively); and *Pseudomonas* was present at similar relative abundance in water (13%), flour (10.1%), and day 1 starters (12%) (Fig. S2A). These shared taxa likely drive the overlap between flour and day 1 starters (Fig. S2B), despite flour-specific differences in bacterial membership (Figs. S2C and S2D). All starter communities followed similar succession patterns with regard to a measure of evenness: both Simpson and Shannon diversity initially increased, reaching a maximum between day 3–6 (Figs. 3A and 3B), then decreased after species of Lactobacillaceae became dominant (Fig. 3C) and pH stabilized (Fig. 2A). Bacterial richness followed similar patterns across all flour types except for rye starters, in which bacterial richness increased dramatically from day 3 to day 6 and retained 15–30 more ASVs, on average, compared to other flour types (Fig. S3).

The observed changes in sourdough starter performance correlated significantly with (and are likely driven by) predictable shifts in the composition of bacterial communities. For example, pH decreased significantly as Lactobacillaceae grew more abundant (Pearson correlation = −0.5321391, $p < 2.2e-16$; Fig. S4). In addition, bacterial community dynamics corresponded with differences in the prevalent aromas produced over time (Fig. 4, Table 1 and Fig. S5). On day 1, when starter communities were dominated by

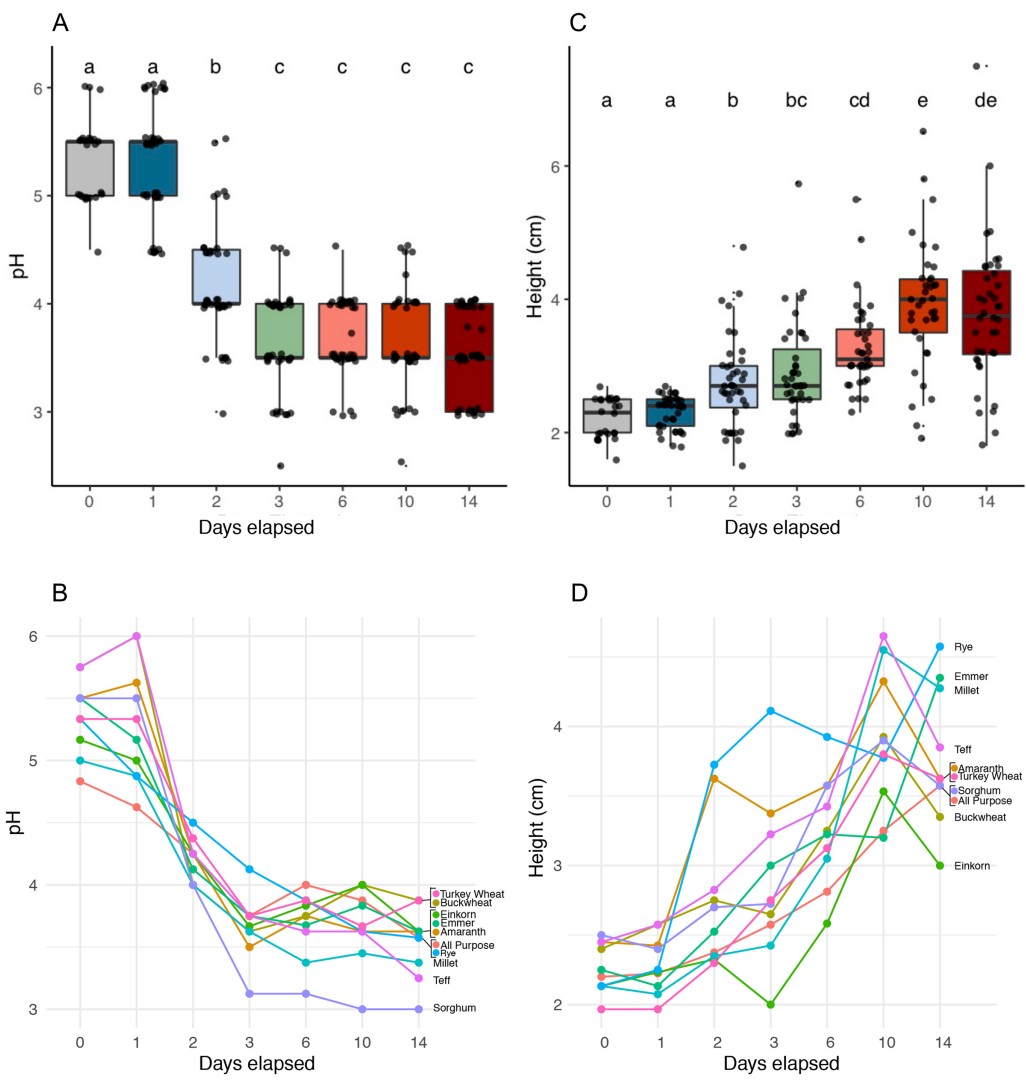

**Figure 2 pH and height (in centimeters) measured for sourdough starters on days 0, 1, 2, 3, 6, 10, and 14.** (A) pH averaged across flour types per time point (Kruskal-Wallis with Bonferroni correction, chi-squared = 158.35, df = 6, $p < 2.2e{-}16$). (B) pH plotted per day for each flour type. (C) Height before mixing (in centimeters), averaged across flour types per time point (Kruskal-Wallis with Bonferroni correction, chi-squared = 121.22, df = 6, $p < 2.2e{-}16$). (D) Height before mixing (in centimeters), plotted per day for each flour type. Lowercase letters above the boxplots denote statistical significance of the pairwise comparisons (Wilcoxon Rank Sum test with Bonferroni correction, with differences significant if $p < 0.05$). Days with the same letter are not significantly different, while days with different letters are significantly different.

bacterial taxa associated with flour and plants, they smelled predominantly of toasted, grain and cereal aromas. Intermediate stages of bacterial succession produced a mixture of sour, grain/cereal, and earthy aromas. However, by day 14, mature starters smelled like sour, fermented fruit (Fig. 4B). When comparing community succession dynamics among starters from day 1–14, PERMANOVA tests confirmed that starter samples harbor significantly different bacterial taxa on day 1, days 2–3, day 6, and days 10–14. When comparing individual taxa between successional days, we detected significant differences in

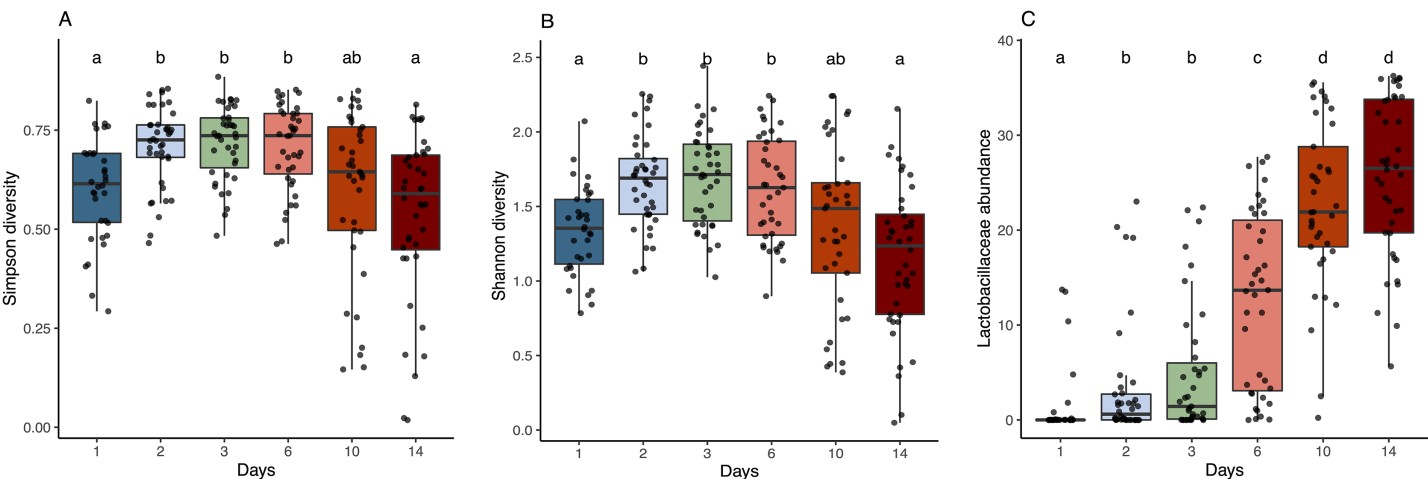

**Figure 3 Alpha diversity and Lactobacillaceae abundance in sourdough starters on days 0, 1, 2, 3, 6, 10, and 14.** (A) Simpson diversity (Kruskal-Wallis with Bonferroni correction, chi-squared = 42.689, df = 5, p = 4.272e−08) and (B) Shannon diversity (Kruskal-Wallis with Bonferroni correction, chi-squared = 38.534, df = 5, p = 2.947e−07) averaged across flour types per time point. (C) Relative abundance of Lactobacillaceae averaged across flour types per time point (Kruskal-Wallis with Bonferroni correction, chi-squared = 143.94, df = 5, p < 2.2e−16). Lowercase letters above the boxplots denote statistical significance of the pairwise comparisons (Wilcoxon Rank Sum test with Bonferroni correction, with differences significant if p < 0.05). Days with the same letter are not significantly different, while days with different letters are significantly different.

the relative abundance of eleven genera including *Erwinia* (Kruskal-Wallis with Bonferroni correction *p* < 2.15e−15), *Pseudomonas* (Kruskal-Wallis with Bonferroni correction *p* < 4.50e−11), *Klebsiella* (Kruskal-Wallis with Bonferroni correction *p* < 1.00e −04), *Pantoea* (Kruskal-Wallis with Bonferroni correction *p* < 2.88e−11), Enterobacteriaceae unkn (Kruskal-Wallis with Bonferroni correction *p* < 3.49e−19), *Weissella* (Kruskal-Wallis with Bonferroni correction *p* < 1.73e−06), *Leuconostoc* (Kruskal-Wallis with Bonferroni correction *p* < 1.03e−04), *Latilactobacillus* (Kruskal-Wallis with Bonferroni correction *p* = 2.52e−03), *Levilactobacillus* (Kruskal-Wallis with Bonferroni correction *p* < 2.45e−04), *Gluconobacter* (Kruskal-Wallis with Bonferroni correction *p* < 4.81e−08), and *Pediococcus* (Kruskal-Wallis with Bonferroni correction *p* < 2.72e−14). *Erwinia* (11.5% on day 1), *Pseudomonas* (12.8–18.2%), *Klebsiella* (11.2–13.2%), *Pantoea* (21.3% on day 1), and Enterobacteriaceae unkn (16.3–28.5%) occurred at higher relative abundance in younger starters (≦3 days old) than in older starters (≧ 10 days old) (Figs. 4 and S5A). Three genera—*Weissella, Leuconostoc*, and *Latilactobacillus*—occurred at higher relative abundance in starters after day 1 but before day 10; and three other genera—*Levilactobacillus, Gluconobacter*, and *Pediococcus*— occurred at higher relative abundance in older starters (≧ 10 days old) (Figs. 4 and S5A).

## Differences among mature starters

Whereas the youngest starters (<3 days; Fig. 4) were characterized by variation both among and within flour types, differences among mature starters (day 14) were dominated by differences among flour types (Table 2, Figs. 5 and S6). These flour-specific differences in bacterial membership coincided with differences in specific aromas (Figs. 5A and S5B). In other words, as starters age, they converge toward specific climax communities

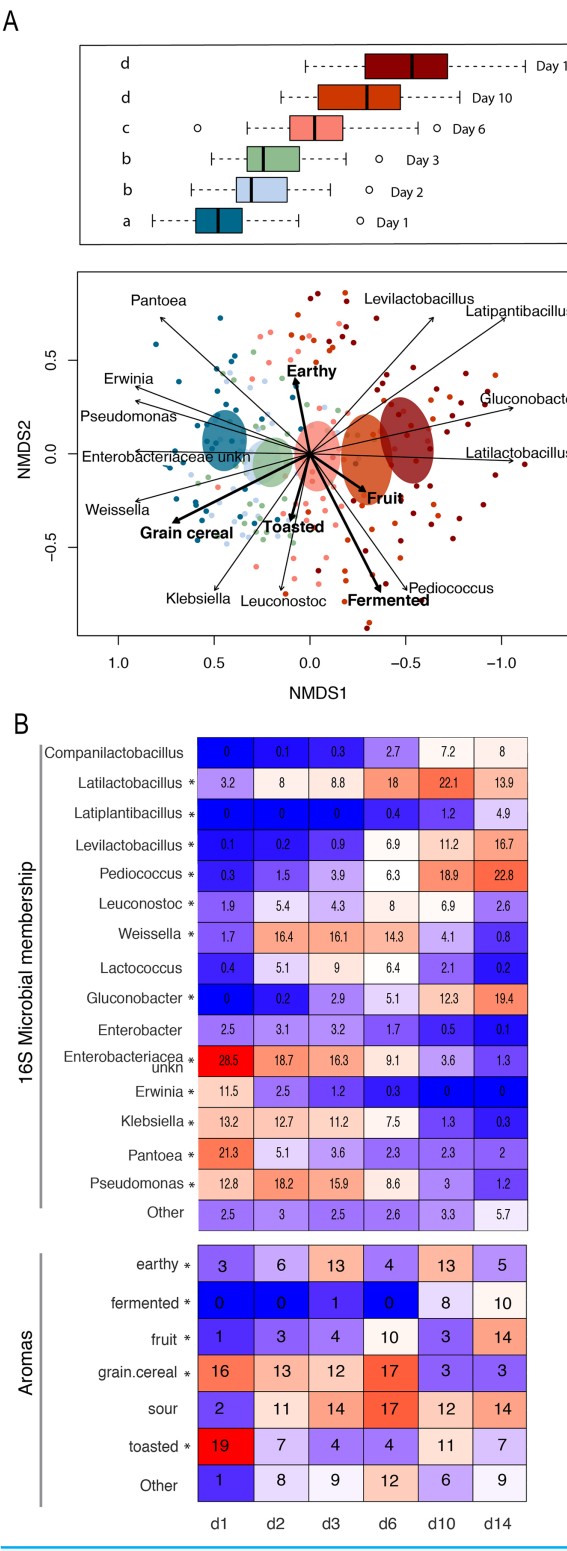

**Figure 4 Patterns of bacterial and functional succession in sourdough starters.** (A) Bray- Curtis NMDS plot of sourdough starter bacterial community composition on days 1, 2, 3, 6, 10, and 14. Box plots in legend show variation in community composition among starters at each time point, with significant pairwise differences ($p < 0.05$ by PERMANOVA) indicated by lowercase letters. Ellipses show the 99% confidence intervals for standard error per time point. Arrows depict the directionality

**Figure 4 (continued)**
(eigenvector) and degree (eigenvalue) of the relationship of each significantly enriched ($p < 0.05$ by Kruskal-Wallis after Bonferroni correction) bacterial taxon or aroma (in bold) to overall community structure. (B) Heatmap of average relative abundance of all bacterial genera present at ≥0.1% relative abundance for each time point sampled and aromas detected in at least 25% (10/40) of the samples. Asterisks (*) denote all taxa and aromas found to be significantly different by day ($p < 0.05$ by Kruskal-Wallis test with Bonferroni correction). Relative abundance (mean +/− standard deviation) for these significant bacterial genera and aromas are shown in Fig. S5.

depending on which flour type they are fed; and different climax communities not only comprise different bacteria but also product different aromas. Two flour types support particularly unique communities: teff starters were dominated by *Pediococcus* (79.2%) and also contained *Leuconostoc*, but lacked genera belonging to the family Lactobacillaceae; and amaranth starters were dominated by *Latilactobacillus* (70.3%, Fig. 5B).

# DISCUSSION

Here we characterized bacterial succession across 14 days (336 h) in sourdough starters grown from 10 different flour types. Given that the generation times of *Lactobacillus* bacteria are roughly an hour (*Venturi, Guerrini & Vincenzini, 2012*), this period provides the opportunity to study succession over more than three hundred generations—a feat that would require over three hundred years for annual plant systems. Across these generations, we found predictable patterns of succession that were similar across flour types, despite large differences in the evolutionary histories and biology of the seeds (and plant lineages) from which these different flours were made. Indeed, succession was far more predictable in the context of these sourdough starters than it tends to be in the forests, grasslands and other macroscopic systems in which the ecology of succession has been best-studied (*Pickett, Collins & Armesto, 1987*). As succession proceeded, the starters underwent two sets of changes that appeared to be deterministic and associated with niche-based processes. In particular, the outcome of succession appears to be influenced by the ability of species to produce and/or tolerate metabolic products, specifically acid; and by competition for resources, with the outcomes of that competition influenced by resources and the traits that favor particular taxa. The genera of bacteria that became dominant as starters matured were not only characteristic of late successional stages but also strongly influenced by flour type, suggesting that the resources available in each grain type favored a subset of lineages best able to compete for those resources.

## Succession processes

Community membership at each successional stage comprised bacteria with specific functional traits: generalists and plant endophytes gave way to lactic acid producing genera, which were outcompeted by a subset of acid producing genera that are also acid-tolerant. Our results are entirely in line with the three-phase model of succession elaborated through the study of wheat-based sourdough (*Ercolini et al., 2013*; *Minervini et al., 2014*; *Harth, Van Kerrebroeck & De Vuyst, 2016*; *Gobbetti et al., 2016*). In the first

**Table 2 Bacterial genera detected in mature (14-day-old) starters grown from 10 different flour types in the current study, compared to bacterial membership detected in previous studies.** Genera from the current study are listed in order of relative abundance; taxa unique to either the current study or to previous studies are listed in bold.

| Flour type | Current study | Previous studies (genus-level) | References |
|---|---|---|---|
| All purpose | Gluconobacter, Levilactobacillus, Pediococcus, Latiplantibacillus, **Loigolactobacillus**, Leuconostoc, **Latilactobacillus**, Companilactobacillus, Pseudomonas, **Enterobacteriaceae unkn** | **Acetobacter**, Companilactobacillus, **Erwinia**, Gluconobacter, **Lacticaseibacillus**, **Lactiplantibacillus**, Levilactobacillus, Leuconostoc, **Limosilactobacillus**, Pediococcus, Pseudomonas | *Landis et al. (2021)* |
| Turkey wheat | Levilactobacillus, **Gluconobacter**, Loigolactobacillus, Lactiplantibacillus, Companilactobacillus, Pediococcus, Latilactobacillus, **Pantoea**, **Pseudomonas** | Companilactobacillus, **Enterococcus**, **Fructilactobacillus**, **Furfurilactobacillus**, **Lacticaseibacillus**, Lactiplantibacillus, **Lactobacillus**, **Lactococcus**, Latilactobacillus, **Lentilactobacillus**, **Lentzea**, **Leuconostoc**, Levilactobacillus, **Limosilactobacillus**, Loigolactobacillus, Pediococcus, **Streptococcus**, **Weissella** | *Landis et al. (2021)*, *De Vuyst et al. (2014)* |
| Emmer | **Levilactobacillus**, **Gluconobacter**, **Companilactobacillus**, **Latilactobacillus**, **Pantoea**, **Pseudomonas**, **Pediococcus**, **Enterobacteriaceae unkn**, Lactiplantibacillus | **Furfurilactobacillus**, Lactiplantibacillus, **Weissella** | *De Vuyst et al. (2014)* |
| Einkorn | **Gluconobacter**, Companilactobacillus, Pediococcus, Levilactobacillus, Lactiplantibacillus, **Leuconostoc**, **Pseudomonas**, **Loigolactobacillus**, **Enterobacteriaceae unkn**, Latilactobacillus, **Pantoea** | Companilactobacillus, Lactiplantibacillus, Latilactobacillus, Levilactobacillus, **Limosilactobacillus**, Pediococcus | *Çakır et al. (2020)* |
| Rye | Levilactobacillus, Companilactobacillus, **Gluconobacter**, **Enterobacteriaceae unkn**, Lactiplantibacillus, **Pantoea**, **Pediococcus**, **Pseudomonas**, **Leuconostoc** | Companilactobacillus, **Fructilactobacillus**, Lactiplantibacillus, **Lactobacillus**, **Lactococcus**, Levilactobacillus, **Limosilactobacillus**, **Weissella** | *Landis et al. (2021)*, *De Vuyst et al. (2014)* |
| Millet | **Gluconobacter**, **Pediococcus**, Levilactobacillus, **Companilactobacillus**, Lactiplantibacillus, **Pseudomonas**, **Enterobacteriaceae unkn**, Latilactobacillus, **Pantoea**, Leuconostoc | **Fructilactobacillus**, **Lacticaseibacillus**, Lactiplantibacillus, **Lactobacillus**, Latilactobacillus, Leuconostoc, Levilactobacillus, **Limosilactobacillus** | *De Vuyst et al. (2014)* |
| Sorghum | **Pediococcus**, **Gluconobacter**, **Latilactobacillus**, **Enterobacteriaceae unkn**, **Leuconostoc**, **Pseudomonas** | **Enterococcus**, **Lactobacillus**, **Lactococcus**, **Limosilactobacillus** | *De Vuyst et al. (2014)* |
| Teff | Pediococcus, Leuconostoc, Latilactobacillus | **Companilactobacillus**, **Fructilactobacillus**, **Lactiplantibacillus**, **Lactobacillus**, Latilactobacillus, **Lentilactobacillus**, Leuconostoc, **Levilactobacillus**, **Limosilactobacillus**, Pediococcus, **Weissella** | *De Vuyst et al. (2014)* |
| Buckwheat | Latilactobacillus, Pediococcus, **Pantoea**, Lactiplantibacillus, **Pseudomonas**, **Gluconobacter**, **Loigolactobacillus**, Levilactobacillus, **Enterobacteriaceae unkn**, Leuconostoc | **Companilactobacillus**, **Fructilactobacillus**, **Lacticaseibacillus**, Lactiplantibacillus, **Lactobacillus**, **Lactococcus**, Latilactobacillus, Leuconostoc, Levilactobacillus, **Limosilactobacillus**, Pediococcus, **Weissella** | *De Vuyst et al. (2014)* |
| Amaranth | Latilactobacillus, **Loigolactobacillus**, Lactiplantibacillus, Pediococcus, Leuconostoc | **Companilactobacillus**, Fructilactobacillus, **Lacticaseibacillus**, Lactiplantibacillus, **Lactobacillus**, **Lactococcus**, Latilactobacillus, Leuconostoc, **Levilactobacillus**, **Limosilactobacillus**, Pediococcus | *De Vuyst et al. (2014)* |

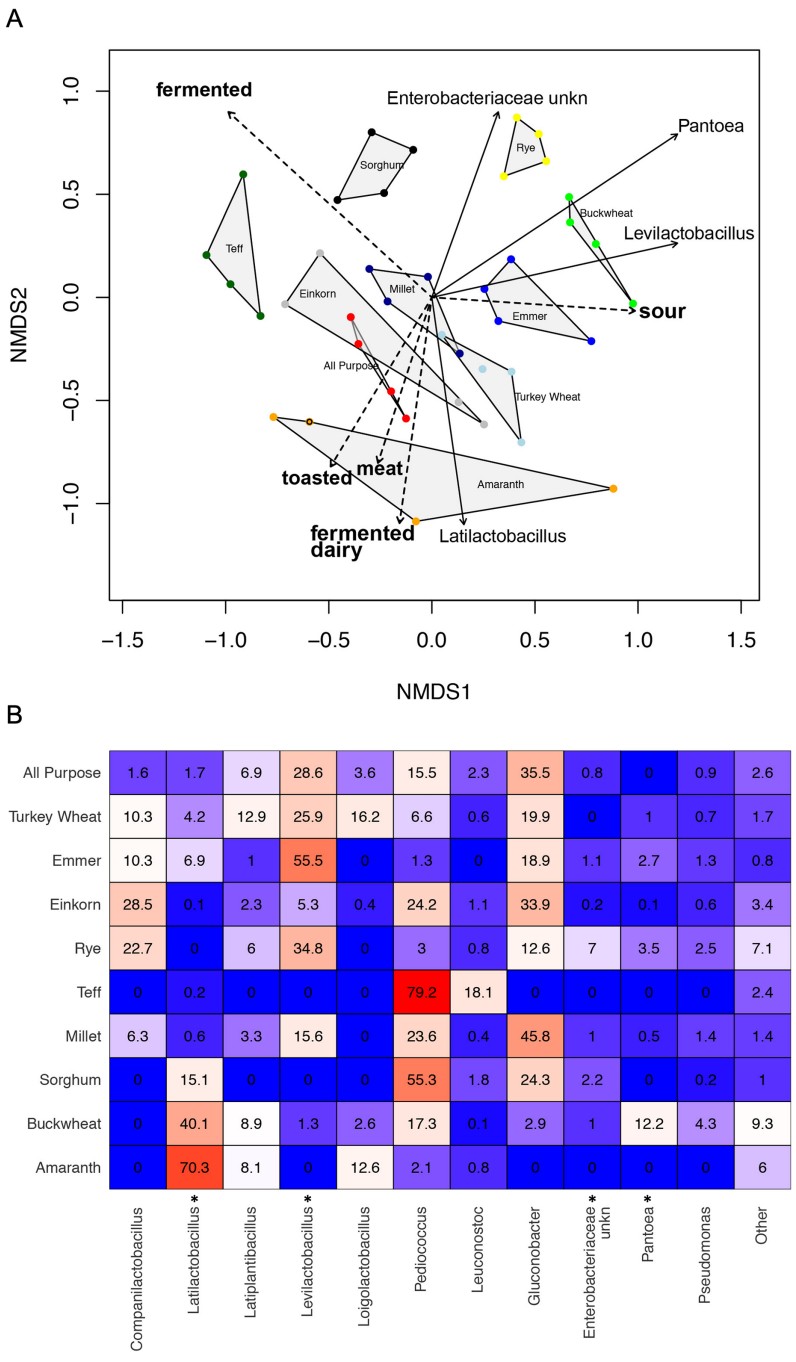

**Figure 5 Comparison of climax community structure and aromas in sourdough starters, measured by flour type on day 14.** (A) Bray-Curtis NMDS plot of sourdough starter bacterial community composition on day 14. Arrows depict the directionality (eigenvector) and degree (eigenvalue) of the relative relationship of each significantly enriched bacterial taxon (solid line; $p < 0.05$ by Kruskal-Wallis after Bonferroni correction) and aroma (dashed line; $p < 0.05$ by envfit permuted regression) to community structure for each flour type. (B) Heatmap of average relative abundance of all bacterial genera present in sourdough starters at ≥0.1% relative abundance for each flour type on day 14. Asterisks (*) denote all taxa found to be significantly different by day ($p < 0.05$ by Kruskal-Wallis test with Bonferroni correction). Relative abundance (mean +/− standard deviation) for these significant taxa are shown in Fig. S7.

phase (day 1) of succession, sourdough starters are both individually diverse (high alpha diversity; Figs. 3A and 3B) and variable one to the next (high beta diversity), even within flour types. We hypothesize that this initial diversity reflects the effects of source pools and which bacterial taxa happen to arrive in a particular starter, as well as the size of the initial populations of each of these bacterial taxa. Ecologists have pioneered a range of approaches to test how important source pools and the relative abundance of different taxa in source pools are to the initial composition of starters (*Miller et al., 2019*; *Reese et al., 2020*). As succession proceeded to phase 2 (days 2, 3, and 6), taxa more characteristic of climax sourdough communities, including LAB, became more abundant and eventually dominated the starters in phase 3 (days 10 and 14). This dominance results in part from the ability of these climax taxa to grow quickly on the available resources and to produce allelopathic compounds, primarily lactic acid, that kill many species (*Minervini et al., 2014*). It is possible that the production of lactic acid also facilitates the arrival of a subset of acidophilic species that thrive in the final successional stage of the sourdough starters (possible but not yet tested). By including the functional outputs (pH, height, and aromas) associated with dominant bacteria, we offer additional nuance beyond the three-phase model, enabling us to characterize a total of 6 statistically distinct functional stages of sourdough succession (Table 1). The dominant aromas detected at each stage correlate with changes in bacterial community composition (Fig. 4), demonstrating that the succession process has measurable functional consequences for the aesthetic properties of sourdough. Indeed, both the bacterial and aromatic profiles follow classic successional shifts, with intermediate stages exhibiting the most diverse and diffuse representation before converging toward climax membership and functionality (*e.g.*, fruit, sour, and fermented aromas).

Our results support several anecdotal observations shared by bakers and participants in our ongoing series of citizen science sourdough projects (http://robdunnlab.com/projects/science-of-sourdough/). For example, many starter-keepers note a dramatic increase in height on day 2 or 3, followed by a plateau or decrease in activity until roughly day 7. We verify this widely noted behavioral pattern across flour types (Fig. 2). We hypothesize that this early peak in starter height, a proxy for gas production, results from the growth of flour-associated yeasts known to dominate young starters (*Ercolini et al., 2013*). In addition, bakers often invoke a "10 day rule", using starters to bake only after they are at least 10 days old. Our results confirm that sourdough starters are functionally mature and stable (*i.e.*, suitable for baking) by day 10 across flour types (Fig. 2), and that this functionality corresponds with dominant bacterial taxa characteristic of sourdough climax communities (Figs. 3 and 4)—specifically, acid-producing genera of Lactobacillaceae (Figs. 3C and S4).

An unexpected result occurred in starters made with rye flour. Starters made with rye flour are very common in some parts of Europe (*Weckx et al., 2010*). However, they have been less well-studied than starters made with wheat flour (*De Vuyst, Van Kerrebroeck & Leroy, 2017*). In general, it has been hypothesized that the lactic acid produced by LAB in mature starters limits the diversity of taxa able to survive in mature starters compared to younger starters (*e.g.*, observations in (*Minervini et al., 2014*; *Gobbetti et al., 2016*)).

The results for all starters supported this prediction except for those made with rye flour, which contained a relatively high diversity of bacterial taxa even after they reached maturity (Fig. S3). Why might rye flour support greater bacterial diversity? Rye contains starches that are easily degraded by amylase compared to more resistant wheat starches (*Perten, 1964*), making sugars more immediately available for microbial metabolism. We speculate that the diversity of nutrients present in whole grain flours, combined with enhanced starch degradation by amylase in rye flour, create additional niche space to allow the persistence of more species in rye starters compared to other grains—a possibility that deserves more exploration.

## Mature starters

Acid producing and acid-tolerant taxa dominate mature starters across all flour types. For example, *Gluconobacter*, *Pediococcus*, and other genera of Lactobacillaceae were most abundant in mature starters. The ubiquity of the genus *Gluconobacter*, which accounted for 12.6–45.8% of the microbial communities present in 7 of 10 flour types studied (Fig. 4), was unexpected. *Gluconobacter* was recently isolated for the first time from sourdough starters in the Henan Province of China (*Xing et al., 2020*) and was also detected in a study of sourdough starters collected from four continents (*Landis et al., 2021*); but the genus has received little attention in studies of starters more generally. Together, these results suggest that acetic acid bacteria may play a greater role in sourdough starters than is currently appreciated.

Climax communities are likely shaped by flour-specific characteristics (Fig. S6), though future research is needed to determine to what extent these climax communities are shaped by flour compounds *vs* priority effects. When considering acidity, dough height, or the relative abundance of LAB, the starters are far more similar within than between different flour types. We also see key differences among flour types, both in the particular bacterial taxa (ASVs) that are present and the aromas associated with those communities (Table 2 and Fig. 5). *Pantoea* is much more abundant in mature buckwheat starters (12.2%) than in other starters ($\leqq 3.5\%$). Acid-tolerant *Leuconostoc* is enriched in teff (18.1%), and an unknown genus of *Enterobacteriaceae* is enriched in rye starters (7%). Other starters were characterized by the absence of particular taxa: for example, *Gluconobacter* is present in all starters except teff and amaranth (Fig. 5B). The relative abundance of *Pediococcus* is negatively correlated with other Lactobacillaceae genera (*Companilactobacillus*, *Latilactobacillus*, *Latiplantibacillus*, *Levilactobacillus*, and *Loigolactobacillus*), suggesting that they may compete for resources. This is especially apparent in teff (*Latilactobacillus* = 0.2%, *Pediococcus* = 79.2%; fermented aromas) and amaranth (*Latilactobacillus* = 70.3%, *Pediococcus* = 2.1%; toasted, meat, and fermented dairy aromas) starters. *Landis et al. (2021)* previously established that different sourdough communities produce distinct aromatic profiles, and the current project suggests that bakers can use different flour types to select for specific bacteria and aromas (Figs. S5 and S6), which in turn impact bread characteristics. Importantly, while we focused on bacteria in this study, yeast communities also contribute to aromatic profiles through direct formation of volatile

compounds (*e.g.*, alcohols, esters, carbonyl compounds (*Hansen & Schieberle, 2005*)) and *via* interactions with LAB (*Guerzoni et al., 2007*).

Many of the bacteria we detected are novel compared to previous studies of the same flour types (Table 2). These differences might indicate that DNA sequencing removes the LAB bias of culture-based methods, enabling us to better appreciate the diversity of sourdough taxa; or they may result from regional differences among grains and/or bacterial species pools. Our study design does not allow us to distinguish why the taxa present in the climax communities differed among flour types. Cereal grains differ in many respects, including enzyme activities, phenolic compounds, and buffering capacity, which have been shown to impact sourdough microbiota in previous studies (*Sekwati-Monang, Valcheva & Gänzle, 2012*; *Dinardo et al., 2019*). Another possibility is that nutritional differences among flour types may favor specific taxa and bacterial interactions, yielding predictable community structures that perform particular functions as previously described (*Landis et al., 2021*). Given that sourdough microbes can reflect grain phenology (*Minervini et al., 2015*) and cultivation (*Rizzello et al., 2015*), future efforts to compare starters grown from different flour batches (*i.e.*, the same cereal grown in different fields or years) could yield additional insights to the variation within *vs* among cereals.

## Broader context

The successional patterns characterized here are similar to those previously observed in other lacto-fermented foods. Most spontaneously lacto-fermented foods (*e.g.*, sourdough starter, sauerkraut, and kimchi) take only 2 weeks to develop stable climax communities. Chinese sauerkraut (*Xiong et al., 2012*) and dongchimi—a traditional Korean kimchi made with radishes (*Jeong et al., 2013*)—both undergo undisturbed fermentation for 10 days, during which time LAB levels increase and pH drops to ≤4 and taxa shift from opportunistic generalists including *Enterococcus faecalis* (*Xiong et al., 2012*), *Pseudomonas* and *Weissella* (*Jeong et al., 2013*), to acid-producing climax communities dominated by *Leuconostoc* (*Jeong et al., 2013*), *Lactiplantibacillus plantarum* and *Lacticaseibacillus casei* (*Xiong et al., 2012*). Despite reaching a similar climax community of acid-resistant LAB, these fermentations all differ from sourdough starters in that they don't involve refreshment. It would be interesting to test how succession might proceed in sourdough starters were they not replenished (with flour). We suspect that failing to feed sourdough starters prevents them from developing their characteristic climate communities. If this were the case, the question is why. One possibility is that cabbage and radish provide a large inoculum of LAB, thereby preventing the establishing of opportunistic bacterial species (*Miller et al., 2019*). Another possibility is that the diverse complex carbohydrates in cabbage and radish, combined with the salt employed in their fermentation, prevent opportunistic generalists from dominating (through resource diversity effects (*Pickett, Collins & Armesto, 1987*)) and disfavor species that are not salt tolerant (*Yang et al., 2020*).

## CONCLUSIONS

Together, our results confirm that bacterial communities in sourdough starters follow predictable patterns of succession driven by resource availability and relative niche breadth
over time. Changes in community composition coincided with changes in pH as well as height and aromas, offering empirical support for common baking practices (*i.e.*, the "10-day rule"). Yet, while the general patterns of succession hold across starters, bacterial community composition and aromas in mature starters differed significantly by flour type and are likely driven by grain-specific bacterial and nutritional components, suggesting that bakers (like brewers) can manipulate inputs to produce specific bread characteristics. To this point, we call for more thorough investigations into the sensory qualities of sourdough starters grown from different flours, and of the breads baked from those starters. A comparison of the aromas detected by a trained sensory panel *vs* the general public (*e.g.*, home bakers) would be particularly helpful to validate citizen science and other broad applications.

The broad successional patterns we have documented, as well as the differences among climax sourdough communities specific to each flour type, are potentially very useful to bakers. For example—we recorded the lowest pH in sorghum, teff, and millet starters, indicating that they would produce the most tangy/sour loaves, whereas buckwheat or turkey wheat would taste less acidic (Fig. 2C) and amaranth starters might contribute unique meaty, fermented dairy and toasted aromas to bread (Fig. 5A). Regarding potential leavening power, rye, emmer, and millet starters achieved the highest rise, whereas buckwheat and einkorn might be better suited to flatter baked goods (Fig. 2D). As for the "10 day rule": teff, amaranth, sorghum, buckwheat, millet, and einkorn starters reached peak height on day 10, while starters grown from rye, emmer, and all purpose didn't reach peach height until day 14 (Fig. 2D). Our findings enable bakers to know the relative amount of time required to establish a mature sourdough starter, contingent on flour type; and they offer predictions as to which flour type might be used to favor particular microbes or aromas. We were able to reveal these patterns while working with materials readily available to teachers and bakers.

## ACKNOWLEDGEMENTS

We thank Dr. Anne Madden and Leonora Shell, who helped to develop the citizen science protocols that laid the groundwork for this project. Jessie Francese and Remi Wingo implemented the first Sourdough for Science effort at Exploris Middle School, and offered valuable feedback that helped us to scale, simplify, and streamline the protocol. We also thank Boulted Bread bakery, and Joshua Bellamy in particular, for providing flour and baking insights that made this science hands-on and personally relevant. Finally, we thank Martha Calvert for providing constructive feedback on an earlier draft of this manuscript.

### Funding

This work was funded by NSF grant #1319293. The funders had no role in study design, data collection and analysis, decision to publish, or preparation of the manuscript.

## Grant Disclosures

The following grant information was disclosed by the authors:
NSF: #1319293.

## Competing Interests

The authors declare that they have no competing interests.

## Author Contributions

- Erin A. McKenney conceived and designed the experiments, performed the experiments, analyzed the data, prepared figures and/or tables, authored or reviewed drafts of the article, and approved the final draft.
- Lauren M. Nichols conceived and designed the experiments, performed the experiments, analyzed the data, prepared figures and/or tables, authored or reviewed drafts of the article, and approved the final draft.
- Samuel Alvarado performed the experiments, analyzed the data, authored or reviewed drafts of the article, and approved the final draft.
- Shannon Hardy performed the experiments, authored or reviewed drafts of the article, and approved the final draft.
- Kristen Kemp performed the experiments, authored or reviewed drafts of the article, and approved the final draft.
- Rachael Polmanteer performed the experiments, authored or reviewed drafts of the article, and approved the final draft.
- April Shoemaker performed the experiments, authored or reviewed drafts of the article, and approved the final draft.
- Robert R. Dunn conceived and designed the experiments, authored or reviewed drafts of the article, and approved the final draft.

## DNA Deposition

The following information was supplied regarding the deposition of DNA sequences:

The 16S amplicon sequences are available in GenBank: PRJNA973060; accessions SAMN35108252–SAMN35108524.

The raw data are available at Dryad: McKenney, Erin et al. (2023). Data for: Sourdough starters exhibit similar succession patterns but develop flour-specific climax communities [Dataset]. Dryad. https://doi.org/10.5061/dryad.bk3j9kdh3.

## Data Availability

The 16S taxonomic table, aroma data and code are available in the Supplemental Files.

## Supplemental Information

Supplemental information for this article can be found online at http://dx.doi.org/10.7717/peerj.16163#supplemental-information.

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
