# Peer review of "Sourdough starters exhibit similar succession patterns but develop flour-specific climax communities"

_PeerJ, doi:10.7717/peerj.16163_

## Round 0.1 · original submission · Major Revisions

Dear authors, I apologize for the delay in getting back to you with reviewer reports. As you can see, both reviewers see merit in your work and are overall supportive. Both however have pointed out several shortcomings, with which I overall agree.

In brief, I would like to ask you to pay attention to
1) Level of presented detail and rationale of your data analyses to permit reproducibility (including a potential supplementary figure to make the overall workflow more easily accessible) along with basic sequencing results,
2) Carefully revising/toning down statements regarding functionality (only metabarcoding data are provided, no data on functional gene content or metabolite profiles). Here it would be helpful to provide a detailed description in the methods of what ‘functional outputs’ entail in this context – while the study was designed with baking applications in mind, a proportion of the readers will be microbiologists and/or bioinformaticians, for which the term ‘functionality’ in this context might be confusing.
3) Elaborating on bacterial (introduction) and yeast communities (discussion) to aromatic profiles of sourdough starters.

I hope you will be able to prepare a revision of your manuscript accordingly.

Warm regards,

Reviewer 1 ·

Basic reporting

General comments:
In this manuscript, research questions are well-defined, relevant, meaningful, and original. This research fills an identified knowledge gap. Conclusions are well-stated and linked to the original research question. I, therefore, recommend this manuscript to be published with minor revisions, after the authors address the following questions and concerns. Also, I recommend checking the manuscript’s format before publication.

Specific Comments:
Abstract:
I suggest that the authors should add “keywords” at the end of the Abstract.
Introduction:
Generally, the introduction shows context and literature well referenced and relevant. The authors need to give readers some background information about bacteria-related aromas/flavors in the Introduction section.
L43-44: “a variety of bacteria and yeasts with diverse origins ….”. The names of these representative bacteria or yeasts should be given here. In L231, authors the “lactotax” tool to reclassify all ASVs classified as Lactobacillus, suggesting this genus is important. If the genus Lactobacillus plays an important role in sourdough starters, an extra introduction about the role of the genus Lactobacillus is needed.
L115: Consider revising “CO2” to “CO2”.

Materials and Methods:
In general, a better writing of Materials and Methods is required to make this section more logical and coherent. Also, more information is needed to clarify the methods. Some website citation links cannot be opened. I also recommend that authors include all the codes in supplementary materials.
L193: Consider revising “20C” to “20°C”.
L217: It is not clear which Illumina platform was used.
L218: “515f/806r for bacteria”. The primer sequences should be given here.
L221-223: Were DADA2 pipelines performed in R? If so, authors should include the R script of DADA2 analysis as a part of supplementary materials. Also, the website link cannot be opened.
L226: The website link for reviewers cannot be opened.
L230-232: The reason why the genus Lactobacillus was reclassified is not clear. Are there any advantages of the “lactotax” tool? Or are there any particular interests of the genus Lactobacillus?"
L240-241: Consider revising “downstream analysis” to “downstream analyses”. It is not clear what the downstream analyses are. Do downstream analyses include all alpha and beta diversity analyses (e.g., ASV richness, Simpson diversity, metaMDS, etc.)?

Results:
In general, the results are solid and statistical analyses are robust.
L265: I advise authors to add another paragraph before “Characterizing functional stages of succession”. This paragraph should be focused on the Illumina sequencing results. The authors mentioned that this dataset was rarefied to 1,200 bacterial reads per sample. Authors should also report basic sequencing results and qualities such as the number of raw reads after sequencing, the number of reads after quality control, etc.

Discussion:
Some suggestions: this manuscript shows us a good heatmap (Figure 3B), which indicates the relationship between dominated bacteria and aromas. More discussion about this relationship should be added to the Discussion section. The authors should also give readers some background information about bacteria-related aromas/flavors in the Introduction section.
Figures and Tables
Supplementary Figure 6 (“peerj-83903-SupplementaryFigure6”): This figure is not clear. A high-resolution plot is needed.

Experimental design

no comment

Validity of the findings

In general, the results are solid and statistical analyses are robust.

Annotated reviews are not available for download in order to protect the identity of reviewers who chose to remain anonymous.

Reviewer 2 ·

Basic reporting

1. The quality of the figures needs to be improved. The images appear to be low-resolution, resulting in unclear visibility of many data points. Additionally, the legends for the figures should provide more informative descriptions.

2. There is no legend description for Supplementary Figures. This information should be provided.

3. A workflow figure for study design is needed in Figure 1 or Supplementary Figure 1.

Experimental design

1. To ensure the reproducibility of the study, it is essential to include detailed information regarding the methods and protocols used. However, the authors did not provide an adequate description of the data analysis or the characterization of functional stages in microbial succession.

2. The statistical methods utilized in the data analysis should be further elaborated upon. It would be beneficial to include the rationale behind the selection of specific statistical tests and provide references where appropriate. For example, if proper correlation analysis was not conducted in the study, it is not appropriate to use the term 'correlate' in reference to the study. Multiple testing correction should be conducted and reported in the manuscript.

Validity of the findings

The raw data deposit in the BioProject PRJNA973060 is not currently publicly available.

Additional comments

Details are listed in the PDF attachment!

Annotated reviews are not available for download in order to protect the identity of reviewers who chose to remain anonymous.

---

## Round 0.2 · Minor Revisions

Both reviewers and I are pleased with the thorough revision of your manuscript.

As you see, reviewer 1 has pointed out some minor issues which should be addressed prior to acceptance.

1. Please clearly state whether specific pH, height, or elapsed days were used to determine the successional stages.

2. Kindly further clarify the method used for removing contaminants from sequencing data (sequencing of PCR controls and bioinformatic tool).

3. Please refer to the reviewer's final additional comments regarding minor editing issues.

Reviewer 1 ·

Basic reporting

Dear Editor,

The manuscript has been significantly improved after the revision. The authors have addressed all my questions and concerns. Therefore, I recommend that this manuscript should be formally accepted and published.

Experimental design

NA

Validity of the findings

NA

Additional comments

NA

Reviewer 2 ·

Basic reporting

The authors have addressed most of the reviewers' comments. However, certain comments need additional clarification before the manuscript can be accepted for publication.

Experimental design

1. In the section titled 'Characterizing Functional Stages of Microbial Succession,' although the authors have clarified that 'We graphed the pH and height of each starter using ggplot2 in R (R Core Team, 2019) to identify distinct stages of succession across time and flour types in relation to starter performance (i.e., production of acid, CO2, and aromas),' it remains unclear how the functional stages were characterized. This is because ggplot2 is a function in R primarily used for creating visualizations, rather than a statistical method. The authors are advised to clarify whether specific pH, height, or elapsed days were used to determine the different stages. It would be beneficial to include the rationale behind this choice and provide references where appropriate.
2. Regarding the analysis of the 16s rRNA sequence, the authors need to further clarify the method used for removing contaminants. It is recommended that the authors specify whether they sequenced PCR controls or environmental samples in order to effectively address contaminants during the samples prep and sequencing.

Validity of the findings

The authors should make sure that BioProject PRJNA973060 is open to the public before the publication.

Additional comments

1. Inserting images into PDF/Word documents may reduce the resolution of figures. Authors are advised to consult the editor to resolve this issue before publication.
2. 324-326: ‘For example, pH decreased significantly as Lactobacillaceae grew more abundant (Pearson correlation = -0.5321391; Supplementary Figure 4).’ The P-value should also be included.
3. 24-25: ‘We analyzed 16S rRNA sequences using DADA2 and R to assess the bacterial community structure and performance of 40 starters grown from 10 types of flour over 14 days, and identified 6 distinct stages of succession.’ The ‘R’ should be removed as DADA2 is a R script.

---

## Round 0.3 · accepted · Accept

Thank you for the thorough revision.